# “Force-From-Lipids” Dependence of the MscCG Mechanosensitive Channel Gating on Anionic Membranes

**DOI:** 10.3390/microorganisms11010194

**Published:** 2023-01-12

**Authors:** Yoshitaka Nakayama, Paul R. Rohde, Boris Martinac

**Affiliations:** 1Molecular Cardiology and Biophysics Division, Victor Chang Cardiac Research Institute, Sydney 2010, Australia; 2Faculty of Medicine, St Vincent’s Clinical School, The University of New South Wales, Sydney 2010, Australia

**Keywords:** bacterial mechanosensing, bacterial electrophysiology, patch fluorometry, micropipette aspiration, membrane stiffness, *Corynebacterium glutamicum*, MscCG, MscL, MscS, monosodium glutamate (MSG), phosphatidylglycerol (PG)

## Abstract

Mechanosensory transduction in *Corynebacterium glutamicum* plays a major role in glutamate efflux for industrial MSG, whose production depends on the activation of MscCG-type mechanosensitive channels. Dependence of the MscCG channel activation by membrane tension on the membrane lipid content has to date not been functionally characterized. Here, we report the MscCG channel patch clamp recording from liposomes fused with *C. glutamicum* membrane vesicles as well as from proteoliposomes containing the purified MscCG protein. Our recordings demonstrate that mechanosensitivity of MscCG channels depends significantly on the presence of negatively charged lipids in the proteoliposomes. MscCG channels in liposome preparations fused with native membrane vesicles exhibited the activation threshold similar to the channels recorded from *C. glutamicum* giant spheroplasts. In comparison, the activation threshold of the MscCG channels reconstituted into azolectin liposomes was higher than the activation threshold of *E. coli* MscL, which is gated by membrane tension close to the bilayer lytic tension. The spheroplast-like activation threshold was restored when the MscCG channels were reconstituted into liposomes made of *E. coli* polar lipid extract. In liposomes made of polar lipids mixed with synthetic phosphatidylethanolamine, phosphatidylglycerol, and cardiolipin, the activation threshold of MscCG was significantly reduced compared to the activation threshold recorded in azolectin liposomes, which suggests the importance of anionic lipids for the channel mechanosensitivity. Moreover, the micropipette aspiration technique combined with patch fluorometry demonstrated that membranes containing anionic phosphatidylglycerol are softer than membranes containing only polar non-anionic phosphatidylcholine and phosphatidylethanolamine. The difference in mechanosensitivity between *C. glutamicum* MscCG and canonical MscS of *E. coli* observed in proteoliposomes explains the evolutionary tuning of the force from lipids sensing in various bacterial membrane environments.

## 1. Introduction

Mechanosensitive (MS) ion channels are ubiquitous biological force sensors found in organisms from all kingdoms of life [1,2,3,4]. MS channels are indispensable for biological processes such as osmoregulation [5,6,7], cytokinesis [8,9], gravity sensing [10,11], touch sensation [12,13], and hearing [14,15]. In Corynebacteria, mechanotransduction plays a central role for glutamate export in processes ranging from osmoregulation to microbiota-gut-brain-axis [16]. Due to an exceptionally efficient glutamate synthesis, *Corynebacterium glutamicum* has been utilized for many decades in industrial scale monosodium glutamate (MSG) production. The use of *Corynebacterium glutamicum* as a cell factory for MSG production relies on the activation of evolutionary specialized MscCG-type mechanosensitive channels in bacterial cell membranes to release L-glutamate into culture media [17,18,19]. The model proposed for the MscCG channel activation by increased membrane tension [19,20] is based on the gating model of prokaryotic MscS-like channels of small conductance, which have been shown to sense membrane tension in lipid bilayers according to the “Force-From-Lipids” principle [21,22,23].

To understand the gating mechanisms of MscCG-type channels, bacterial giant spheroplasts of *E. coli*, *Bacillus subtilis*, and *C. glutamicum* have been used to characterize their biophysical properties [17,24,25,26,27,28]. In patch-clamp recordings from *C. glutamicum* giant spheroplasts, negative pressure was applied to patch pipettes to stretch spheroplast membrane patches to activate MS channels (Figure 1A). Two different types of MscS-like channels (MscCG and MscCG2), as well as a MscL-like channel (CgMscL), were recorded in *C. glutamicum* spheroplasts, (from strain ATCC13869) [17]. Importantly, MscCG channels in the *C. glutamicum* native membrane were activated at a lower negative pressure compared to the channels heterologously expressed in cell membranes of other bacteria, e.g., *E. coli*, suggesting the importance of the membrane lipid environment for the channel mechanosensitivity. For bacterial MS channels, it is well-known that the lipid bilayer and protein–lipid interactions are critical for MS channel gating according to the “Force-From-Lipids” principle [21,22,23,29], and that the mechanosensitivity of bacterial MS channels is strongly influenced by the type of lipids within the membrane. Since bacteria lack a cytoskeleton acting as force-transmitting tethers, as in eukaryotic cells, their lipid bilayer is the main tension-bearing element transmitting the force activating MS channels [30]. CryoEM studies of the canonical *E. coli* MscS channel revealed that lipid molecules are bound to the channel in the closed state, and a small N-terminal helix functions as an anchor in the closed-to-open confirmational changes [31]. Activation by de-lipidation with cyclodextrins elucidated further the importance of bound lipids for the MscS gating cycle [32,33]. However, the gating model of *E. coli* MscS may not be sufficient to explain the gating of the structurally more complex MscS-like channels. For example, the Arabidopsis mitochondrial MscS-like channel, AtMSL1, does not have an N-terminal anchor helix contributing to the channel mechanosensing as in the case of *E. coli* MscS [34,35]. Further, the paralogous *E. coli* MscS-like channel, YnaI, has a different gating mechanism with a flexible pore helix [36] in which specific binding of cardiolipin (CA) and phosphatidylserine (PS) was suggested to play a role [37]. Another paralogous channel, MscK of *E. coli*, has a large N-terminal extension that contributes to a curved channel structure and a possible gating mechanism by channel flattening and expanding [38], similar to the Piezo1 channel [39].

As bacterial membrane lipids are highly diverse [40,41], bacterial MS channels would have required the evolution of varied mechanisms for force sensing in their corresponding membrane lipid environments. MscS-like channels, which includes MscCG, are structurally diverse by possessing varied structural domains [42,43]. These channels do not all necessarily function as simple osmotic emergency valves but seem to have more specialized functions [44]. Corynebacterial cell membranes are unique by exclusively consisting of negatively charged lipids, including phosphatidylglycerol and cardiolipin, inferring that the “Force-From-Lipids” gating of MscCG channels may differ from the gating of *E. coli* MscS.

MscCG of *C. glutamicum* was previously studied by patch clamp on giant spheroplast preparations from this bacterium [17]. In this study, we report on the purification of the MscCG channel protein for liposomal reconstitution and its subsequent single channel recording by patch clamp. This reductionist approach was used for a better understanding of MscCG’s inherent mechanosensitivity and the effects that membrane lipid components may have on its gating. Our data demonstrate that MscCG mechanosensitivity is highly dependent on anionic membrane lipids, which present the main structural components of the *C. glutamicum* cell membrane.

## 2. Materials and Methods

### 2.1. Strain, Construct, and Cell Culture

*Corynebacterium glutamicum* ATCC13032 (Kyowa) wild-type (WT) strain was used in this study. The original expression vector pEKex2 was gifted from Prof Reinhard Kraemer’s laboratory [45]. The P_tac_ promotor of pEKex2 was modified to a synthetic promotor from the *C. glutamicum* synthetic promotor library [46] to the intermediate strength constitutional promotor, I29, to create the expression plasmid pEKex2-I29, which allowed optimum overexpression for growth and yield. All *C. glutamicum* cells were cultured with Brain Heart Infusion (BHI) media (BD BBL, 211059) supplemented with 200 mM L-glutamic acid (Sigma-Aldrich, G1251, St. Louis, MO, USA), pH adjusted to 7.6 with KOH (Sigma-Aldrich, 484016) and 250 mM KCl (Sigma-Aldrich, 12636). Colonies were isolated with the above media also containing 2% final Davis Agar (Bacto Laboratories, Sydney, Australia, DAVAGA), 25 μg/mL kanamycin sulphate (Astral Scientific, Gymea NSW, BIOKB0286) and incubated at 31.5 °C, as were starter cultures (with 6 μg/mL kanamycin used for liquid media). Expression cultures for purification were incubated within tissue paper covered flasks at 11 °C, 80 rpm (50.8 mm stroke) from 1/100 starter culture incubation, until OD600 2.5–3.0 and then stored at 4 °C for 40–48 h with additional 250 mM NaCl.

### 2.2. Protein Expression and Purification

Cells were harvested by centrifugation (5000× *g*, 30 min, 4 °C), and the pellets were resuspended in cold phosphate-buffered saline (PBS; 10 mM Na_2_HPO_4_, 1.8 mM KH_2_PO_4_, pH 7.5, 137 mM NaCl, 2.7 mM KCl) supplemented with 500 mM betaine (Sigma-Aldrich, 61962). After adding a tip of spatula of DNaseI (Sigma-Aldrich, DN25) and 175 μg/mL phenylmethylsulfonyl fluoride (PMSF; Roche, 10837091001), cells were processed twice by French press (Thermo Scientific, Waltham, MA, USA) at 20,000 PSI. Cell debris was removed by centrifugation at 14,000× *g* for 30 min at 4 °C, and the supernatant was collected for ultracentrifugation (43,000 rpm in a Type 45 Ti rotor (Beckman Coulter, Brea, CA, USA) for 3 h at 4 °C) to collect the cell membrane. The pelleted membranes were resuspended in PBS with 50 mM betaine at 25 mL/g. An analytical volume was removed for native membrane vesicle liposomal patch-clamp analysis, and the rest was solubilized with 1% (wt/vol) CYMAL5 (Anatrace, C325S, Maumee, OH, USA) overnight at 4 °C. Solubilized membranes were clarified by centrifugation at 14,000× *g* for 30 min at 4 °C. A further analytical volume was taken for membrane protein extract for patch-clamping analysis, whilst the main volume was then incubated with TALON^®^ cobalt affinity resin (Clontech, 635503, Mountain View, CA, USA) for 2 h at 4 °C for binding in the presence of 5 mM imidazole (Sigma-Aldrich, 56750). The resin was gravity column washed with PBS, 50 mM betaine, 5 mM imidazole, 0.4% CYMAL5, 0.1 mg/mL soy polar lipid extract (Avanti, 541602P, Birmingham, AL, USA), and eluted in the same buffer with 200 mM imidazole. After concentrating with Amicon^®^ Ultra-15 100 kDa centrifugal filter unit (Millipore UFC910024), the sample was loaded onto a Superose^®^ 6 10/300 increased column (GE Healthcare) for size exclusion chromatography. Fractions of interest were pooled and concentrated with a Vivaspin^®^2, 2 mL 100 kDa centrifugal concentrator (Cytiva, Marlborough, MA, USA) for gel analysis and liposomal reconstitution.

### 2.3. Proteoliposome Preparations Containing Reconstituted Purified Protein

The purified MscCG channel proteins were reconstituted into liposomes using a Dehydration/Rehydration (D/R) reconstitution method [47] with minor modifications. Ten mg of soy polar extract (Avanti: 541602) or *E. coli* polar extract (Avanti, 100600) was dissolved in chloroform in a glass tube. The yellowish lipid solution was dried with an N_2_ gas jet, and the lipid film was resuspended with D/R buffer (200 mM KCl, 5 mM HEPES adjusted to pH 7.2 with KOH) at 10 mg/mL. To obtain a fine liposome suspension, vortex mixing and bath sonication for 15 min were applied until the suspension became adequately cloudy. Two hundred µL of 10 mg/mL aliquots were used for a proteoliposome preparation. Purified protein was added to the lipid aliquot at desired wt/wt ratio (MscL: lipids = 1:1000, MscS: lipids = 1:250, and MscCG: lipids = 1:100), incubated overnight at 4 °C for incorporation. Bio-Beads™ (Bio-Rad, SM-2, Hercules, CA, USA) were added and incubated with the sample for 3 h at room temperature to remove detergents. The supernatant was transferred without Bio-beads for ultracentrifugation at 43,000× *g* rpm in a Type 50.2 Ti rotor (Beckman Coulter) for 30 min. at 4 °C. Pelleted liposomes were resuspended with 60 µL of D/R buffer and spotted on a microscopic slide. The resuspended liposomes were dehydrated overnight in a vacuum desiccator and rehydrated with D/R buffer for 3 h to overnight for patch-clamp recordings.

### 2.4. Liposome Preparations Fused with Corynebacterial Membrane Vesicle or Reconstituted with Membrane Protein Extract

20 μL collected membrane vesicle samples (as described above) were mixed with 200 µL of 10 mg/mL soy polar extract azolectin liposomes in D/R buffer (200 mM KCl, 5 mM HEPES-KOH, pH 7.2) for a 50:50 ratio and were mixed for 1 h at room temperature before being spotted onto a microscopic slide. The samples were vacuum-desiccated overnight followed by rehydration with 40 μL D/R buffer for 3 h to overnight for patch-clamp recordings. The membrane protein extract aliquot (described above) was added to 200 µL of 10 mg/mL soy polar lipid extract in D/R buffer, and detergent was removed with Bio-Beads™. Dehydration-rehydration was conducted as described above for patch-clamp recordings.

### 2.5. Patch Clamp Electrophysiology

A small aliquot (5 μL) of rehydrated proteoliposomes was placed on the bottom of a recording chamber filled with the patch-clamp buffer (200 mM KCl, 40 mM MgCl_2_, 5 mM HEPES adjusted to pH 7.2 with KOH). After 15–30 min incubation for blister formation, unilamellar blisters emerging from the liposomes were used in all recordings. Borosilicate glass pipettes (Drummond Scientific, 2-000-100, Broomall, PA, USA) were pulled with a Narishige puller (PP-83) to obtain patch pipettes with a resistance of 3.0–4.5 MΩ. Symmetrical ionic solutions were used in all recordings. Channel currents were amplified and sampled at 5 kHz with filtration of 1 kHz using an Axopatch™ 200B amplifier (Molecular Devices, San Jose, CA, USA), Digidata^®^ 1440 A, and pClamp™ 10 software. Negative pressure was controlled with a high-speed pressure clamp (HSPC) apparatus (ALA Sciences, Farmingdale, NY, USA) to apply ramp and step pressures to activate mechanosensitive channels.

### 2.6. Micropipette Aspiration Combined with Patch Fluorometry

Fluorescent images of excised patch membranes labelled with 18:1 Liss Rhodamine PE (Avanti: 810150) (0.1%) were observed with a Zeiss LSM700 confocal microscope equipped with a long working distance water immersion objective (63×; NA 1.15; Carl Zeiss, Oberkochen, Germany). A 555-nm laser line was used to excite the fluorophore, and the emission was detected using a long pass 560-nm filter. To visualize liposome patches inside of the pipettes, the pipette tip was bent 28–30° with a microforge (Narishige; MF-900, Amityville, NY, USA) to become parallel to bottom face of the chamber. The same buffer (200 mM KCl, 40 mM MgCl_2_, 5 mM HEPES adjusted to pH 7.2 with KOH) was used for the patch-clamp studies. Pressure steps were applied with a high-speed pressure clamp (HSPC) without applying any voltages. From confocal images taken of the membrane patch, all parameters to calculate the areal elasticity modulus K_A_ were measured, including membrane tension (T) and membrane area (A), as shown in Appendix A [48]. The areal elasticity modulus K_A_ was determined from the slope of the membrane tension vs. areal strain graph using the following equation:$$K_{A}=\Delta T/\alpha$$

The membrane tension T was calculated based on Laplace’s law shown below. R_d_ is the radius of curvature of the membrane patch and P is the pressure applied to the patch membrane with HSPC.
$$T={\mathrm{PR}}_{d}/2$$

The membrane area after the application of pressure A was calculated using the following equations for the conical pipette shape as in our experiments.
$$A=\pi \left(r+R\right)\sqrt{{\left(R-r\right)}^{2}+L^{2}}+2\pi R_{d}h$$

L is the protrusion length. r and R are the pipette radii at the pipette tip and at the dome, respectively. h is the height of the dome of the patch membrane under pressure. The initial membrane area (zero pressure) A_0_ was calculated as below. The patch dome was assumed to be flat without pressure.
$$A_{0}=\pi \left(r+R\right)\sqrt{{\left(R-r\right)}^{2}+L^{2}}+\pi R$$

The areal strain (fractional area change) α was calculated by measuring the relative change in the area of the expanded membrane inside the pipette A to the initial area A_0_. Thus, the areal strain α is defined as below.
$$\alpha =\frac{\;A\;-\;A_{0}}{A_{0}}$$

## 3. Results

### 3.1. Mechanosensitivity Difference between the MscCG Channels Reconstituted in Liposomes Fused with Native Membrane Vesicles and Liposomes Containing Membrane Protein Extract

To evaluate their functionality, the activity of the MscCG channels were overexpressed homologously in native *C. glutamicum* to yield raw membrane vesicles fused with liposomes to record by the patch-clamp technique (Figure 1A). The membrane vesicles were mixed and fused with polar azolectin liposomes at 1:1 (wt/wt) ratio. The resulting proteoliposomes formed large unilamellar blisters in the patch-clamp recording buffer (200 mM KCl, 40 mM MgCl_2_, 5 mM HEPES-KOH, pH 7.2) allowing for the formation of tight giga-ohm seals between the glass wall of patch pipettes and liposomal membranes (1–3 GΩ). Negative saw-tooth pressure ramp was applied at +60 mV pipette voltage in inside-out excised patch mode. Initially, we examined MscCG and CgMscL at their native expression levels in cells as control experiments to examine a possible effect of the MscCG overexpression. When the pressure was gradually increased, first, small MscS-like currents (MscCG) of approximately 20 pA were recorded, followed secondarily by large MSc-like currents (CgMscL) of approximately 180 pA opening at higher pressures (Figure 1B, top). It should be noted that the *C. glutamicum* strain used, ATCC13032 (Kyowa, Tokyo, Japan), expresses only MscCG, but not MscCG2 like other strains that express both genes [18]. In addition, there is only one MscL-like channel, CgMscL, present in the genome of this bacterium. MscCG currents were rarely encountered (3 in 9 patches) and mostly only a few channels (1.7 ± 1.2 channels, mean ± SD, *n* = 3) were recorded. In contrast, CgMscL currents were always encountered (9 in 9 patches) with the number of recorded channels varying from patch to patch (4.0 ± 5.4 channels, mean ± SD, *n* = 9). We also occasionally encountered a large number of CgMscL channels compared to the average number in our recordings (Appendix A). This indicates that natively expressed CgMscL channels may be clustered in *C. glutamicum* membranes, as previously reported for EcMscL [49]. To evaluate the overexpressed MscCG channels in *C. glutamicum*, liposomes fused with membrane vesicles containing the overexpressed channels were examined under the same experimental conditions. Negative pressure applied to a patch pipette reached −33 mmHg, and 18.2 pA currents of 26 MscCG channels activated before CgMscL currents were recorded at −79 mmHg (Figure 1B, bottom). The currents of the overexpressed MscCG channels were present in all patches (15 out of 15 patches), and the recorded channel number was much larger (30.7 ± 22.5 channels, mean ± SD, *n* = 15) compared to natively expressed MscCG (4.0 ± 5.4 channels, mean ± SD, *n* = 9).

Next, we evaluated the impact of membrane solubilization by 1% CYMAL5 on MscCG channel function to examine how the solubilisation of native lipids by detergent would affect the channel properties. The MscCG protein extract was reconstituted into polar azolectin liposomes at the ratio of 1:1 (wt/wt) (Figure 1A). Upon application of the negative pressure ramp to a liposome patch, 20 MscCG channels were activated before CgMscL was activated (Figure 1C). The recorded number of MscCG channels (28.8 ± 27.6 channels, mean ± SD, *n* = 13) was comparable to the channel number in liposomes fused with non-solubilised membrane vesicles, indicating that the overexpressed MscCG channels were fully functional after membrane solubilization with 1% CYMAL5. However, we noticed that the MscCG channels from the reconstituted solubilised membrane protein extract required higher pressure for activation than the channels in liposomes fused with membrane vesicles, whereas CgMscL channels from the solubilised protein extract required similar activation pressure as the channels from the native resuspended membranes (Figure 1C, black and red traces). Boltzmann distribution curves clearly show that the midpoint activation threshold P_0.5_ of MscCG channels in liposomes containing membrane protein extract were right shifted to −77.3 ± 1.8 mmHg (*n* = 6) from −53.8 ± 1.4 mmHg (*n* = 7) recorded with the channels from resuspended membrane vesicles (Figure 1D). Furthermore, Figure 1E shows that the first opening threshold of MscCG in liposomes fused with membrane vesicles was −39.2 ± 8.1 mmHg (mean ± SD, *n* = 15), whilst the threshold in liposomes containing detergent extracted channel proteins was −58.5 ± 8.6 mmHg (mean ± SD, *n* = 13). In comparison, CgMscL had the same activation threshold in both cases, i.e., −72.4 ± 15.4 mmHg (mean ± SD, *n* = 11) and −78.4 ± 10.5 mmHg (mean ± SD, *n* = 9) in liposomes containing vesicles and protein extract, respectively (Figure 1E). The ratio of the opening thresholds between CgMscL and MscCG (P_L_/P_CG_) were calculated as 2.00 ± 0.13 (*n* =11) and 1.37 ± 0.07 (*n* = 9) (mean ± SEM) for vesicles and protein extract, respectively (Figure 1F). These values are statistically different indicating that the MscCG channels from the protein extract remain mechanosensitive, but their activation threshold is higher compared to the channels in liposomes fused with membrane vesicles.

### 3.2. Protein Purification and Liposomal Reconstitution of the MscCG Channel

The MscCG monomer consists of an N-terminal domain with three transmembrane helices (TM1-3) and a large C-terminal extension with a fourth transmembrane helix (TM4) (Figure 2A). The monomeric MscCG size is 533 aa, and its protein mass calculated from its amino acid sequence is 57.1 kDa. In comparison, *E. coli* MscS has three transmembrane helices (TM1-3). Monomeric MscS size is 286 aa and protein mass is 31.2 kDa. Since the MscS channel is a homoheptamer, the oligomerized MscS size is 218.4 kDa. After membrane solubilization, N-terminal 6×histidine-tagged MscCG was purified by immobilized metal affinity (IMAC) and size exclusion chromatography (SEC) (Figure 1A). The SEC profile with a Superose^®^ 6 increase 10/300 column indicated that a peak at 11 mL of the retention volume was a possible oligomerized MscCG size (Figure 2B). However, the 11 mL peak size that was larger than the 669 kDa reference protein Thyroglobulin did not match the assumed homoheptameric MscCG size calculated with its amino acid sequence (57.1 kDa × 7 = 404.7 kDa). On the SDS-PAGE gels, MscCG protein from the 11 mL peak fractions migrated corresponding to a molecular mass of approximately 90 kDa in denaturing conditions (Figure 2C).

The purified protein from the 11 mL peak was reconstituted into liposomes made of 100% soy polar azolectin at the protein:lipid ratio of 1:100 (wt:wt), and its mechanosensitive channel activities were recorded by the patch clamp technique. Negative pressure steps were increased by −10 mmHg and applied to liposome membranes in inside-out excised patch mode. When the pressure reached −70 mmHg, MscCG currents of approximately 20 pA were recorded at +60 mV pipette voltage and additional channel currents were further activated as the pressure increased (Figure 2D). The currents were observed in 28 out of 49 proteoliposome patches while no currents were observed in “empty” control liposomes. The Boltzmann distribution curve fitted to multiple experiments shows that the midpoint activation threshold P_0.5_ of the purified MscCG channel in the proteoliposomes was −81.1 ± 2.3 mmHg (*n* = 7) (Figure 2E). This value is significantly higher than the threshold of MscCG channels recorded in liposomes containing membrane protein extract. To confirm the high activation threshold of the MscCG channel, we collected more recordings to determine the average first opening activation threshold of the channel. From 28 independent patches, the threshold was determined as −71.5 ± 14.4 mmHg (mean ± SD, *n* = 28) (Figure 2F). This activation threshold is comparable to the threshold of activation of *E. coli* MscL, which is very high for a MscS-like channel given that MscS in azolectin liposomes is activated in the range between −20 and −40 mmHg [29].

To understand how the mechanosensitive channel activities of the purified MscCG protein may be affected by the bilayer lipid environment, we conducted functional characterization of the MscCG currents in 100% polar azolectin liposomes. MscCG is known to exhibit strong hysteresis by having different activation thresholds for the channel opening and closing [28]. To examine the MscCG gating hysteresis, a pressure ramp of −100 mmHg/10 s was applied. Pressure for the first opening (P_open_) and the last closing (P_close_) were −58 mmHg and −27 mmHg, respectively (Figure 3A), indicating a pronounced gating hysteresis. Voltage dependency of the channel opening and closing from four independent patches revealed that the opening pressure (P_open_) was not affected by voltage, while the closing pressure (P_close_) was lower at negative pipette voltages larger than −70 mV (Figure 3B), indicating that membrane potential has impact on the channel closing. In previous studies conducted in bacterial spheroplasts [17,28], MscCG was, unlike MscS, shown not to desensitize or inactivate. Consistently, the MscCG currents showed stable gating upon pressure step of −60 mmHg and did not exhibit any desensitization or inactivation at +60 mV pipette potential. However, several different subconducting current levels were detected when −60 mmHg pressure step was applied at +160 mV pipette voltage (Figure 3C) similar to what was reported for *E. coli* MscS [50]. It should be noted that the high voltage of +160 mV did not activate the channel currents by itself, and only negative pressure was able to activate the currents (Figure 3C). MscCG has also been shown to exhibit strong current rectification by being more conductive at positive pipette voltages compared to negative values [17]. Figure 3D shows the first 5–6 channel current steps activated at a −50 mmHg pressure step and voltage ramp from −80 to +80 mV. Currents were significantly larger at positive voltages than negative voltages (Figure 3D). Moreover, we measured the single channel currents at voltages from −120 to +120 mV to generate an I-V curve for determination of the single channel conductance. The I/V plot shows significant current rectification (Figure 3E), where the single channel conductance was calculated as 304 ± 18 pS (mean ± SD, *n* = 7) and 116 ± 3 pS (mean ± SD, *n* = 4) at +60 mV and −60 mV, respectively. All the above observations were highly consistent with the characteristics of the MscCG channel reported previously [17,24,28].

### 3.3. Comparison between MscCG Mechanosensitivity in Liposomes Made of Soy and E. coli Polar Extract Lipids

To understand why the purified MscCG channel protein shows high activation threshold in liposomes made of polar azolectin (soy polar extract) lipids, we co-reconstituted the MscCG channels with *E. coli* MscL (EcMscL) as a reference channel to evaluate MscCG mechanosensitivity. EcMscL half-activation by membrane tension P_0.5_ is 12 mN/m, whereas the tension of 20 mN/m, which approximates the bilayer lytic tension, is required for its full activation in azolectin liposomes [29]. When a pressure ramp was applied to liposome membranes containing purified MscCG and EcMscL co-reconstituted into liposomes, EcMscL was activated before the MscCG as the negative pressure increased and the first opening of MscCG was completely masked by multiple EcMscL channel openings (Figure 4A). As the negative pressure decreased, EcMscL also closed first at approximately −40 mmHg followed by the MscCG closing after all EcMscL channels closed (Figure 4A top). This behaviour was reproducibly observed in three independent patches, confirming that in liposomes made of the soy polar lipid extract the activation threshold of the purified MscCG channels was higher than the EcMscL threshold. In comparison, we also tested the co-reconstitution of *E. coli* MscS and MscL in liposomes made of soy polar lipids. MscS always opened before the first opening of MscL (Figure 4B). This confirmed that the MscCG activation threshold is significantly higher than the threshold of MscS reconstituted in soy polar liposomes.

The *C. glutamicum* cell membrane has unique lipid components [51] (Table 1). They are negatively charged lipids, including phosphatidylglycerol (PG), phosphatidylinositol (PI), and cardiolipin (CA). In comparison, major lipid components of soy polar extract are phosphatidylcholine (PC) and phosphatidylethanolamine (PE), and phosphatidic acid (PA) and phosphatidylinositol (PI) as minor components (Table 1). Given the difference in the lipid composition between the *C. glutamicum* cell membrane and soy polar extract, we hypothesized that MscCG activation threshold is set to match the bilayer properties of the *C. glutamicum* membrane. Therefore, the purified MscCG and EcMscL were co-reconstituted into liposomes made of *E. coli* polar lipid extract, and their mechanosensitivity was examined. When a pressure ramp was applied to liposome membrane patches containing MscCG and EcMscL co-reconstituted into *E. coli* polar extract liposomes, MscCG channels were activated before the EcMscL channels (Figure 4C). The average threshold for the first channel opening of MscCG and EcMscL was calculated from multiple recordings as −26.9 ± 5.9 mmHg and −43.8 ± 9.7 mmHg, respectively (mean ± SD, *n* = 9) (Figure 4D), and the threshold ratio (P_L_/P_CG_) was determined as 1.64 ± 0.1 (mean ± SEM, *n* = 9) (Figure 4E). These results indicate that the lower activation threshold of the purified MscCG channels compared to EcMscL can be restored to correspond to the threshold in the *C. glutamicum* native membrane by adequately changing the lipids in liposome preparations.

### 3.4. Phosphatidylglycerol Is the Main Charged Lipid for Native MscCG Mechanosensitivity

*E. coli* polar lipid extract has been used as a model for bacterial membranes [52]. To identify which lipid component in the *E. coli* polar extract (Avanti) (67% PE, 23.2% PG, 9.8% CA in Table 1) affects MscCG mechanosensitivity the most, we attempted to mimic the *E. coli* polar extract using synthetic lipids, 1,2-Dioleoyl-sn-glycero-3-phosphoethanolamine (DOPE), 2-Dioleoyl-sn-glycero-3-phosphatidyl-glycerol (DOPG), and cardiolipin18:1 (CA18:1). For experimental simplicity, we adjusted the lipid components of the *E. coli* polar extract to 70% PE, 20% PG and 10% CA. First, liposomes made of 70% DOPE, 20% DOPG and 10% CA18:1 were used to test whether MscCG channels behaved similarly as in liposomes made of *E. coli* polar extract. In the liposomes made of the synthetic lipids, several MscCG channels were activated before the first opening of EcMscL upon application of a pressure ramp (Figure 5A, left). In multiple experiments, MscCG always opened before EcMscL, and the threshold ratio (P_L_/P_CG_) was calculated as 1.35 ± 0.06 (mean ± SEM, *n* = 9) (Figure 5B), showing that the MscCG activation threshold was always lower than the threshold of EcMscL in these liposomes. Secondly, to investigate the effect of 10% CA in *E. coli* polar extract on MscCG mechanosensitivity, we replaced 10% CA18:1 with 10% DOPG, thus liposomes made of 70% DOPE and 30% DOPG were used for experiments under the same experimental conditions. Similarly, MscCG was activated before the first opening of EcMscL upon application of the same pressure ramp protocol (Figure 5A, centre). The threshold ratio (P_L_/P_CG_) was calculated as 1.31 ± 0.06 (mean ± SEM, *n* = 6) (Figure 5B), indicating that 10% CA in *E. coli* polar extract did not significantly affect MscCG mechanosensitivity. Lastly, to examine the effect of anionic lipids in *E. coli* polar extract (23.2% PG, 9.8% CA) on MscCG mechanosensitivity, we replaced the anionic components of 20% DOPG and 10% CA18:1 with overall neutral 30% DOPC. Thus, liposomes made of 70% DOPE and 30% DOPC were used. In contrast to the other two charged liposome preparations, the first activation of MscCG was very close to the activation of EcMscL, and occasionally MscCG was activated after the EcMscL first opening (Figure 5A, right). In contrast, from the co-reconstitution in 100% soy polar extract liposomes (Figure 4A), many EcMscL channels did not mask the MscCG first opening, which enabled us to determine the threshold ratio (P_L_/P_CG_) as 0.94 ± 0.02 (mean ± SEM, *n* = 7) (Figure 5B). Altogether, these findings indicate that the anionic lipid components (PG and CA present in *E. coli* polar extract) are necessary and sufficient to reproduce MscCG mechanosensitivity in the native *C. glutamicum* cell membrane.

### 3.5. Anionic Membranes Composed of Phosphatidylglycerol Are Softer than Neutral Membranes

In the previous study with *C. glutamicum* giant spheroplasts, micropipette aspiration techniques demonstrated that *C. glutamicum* cell membranes are much softer than *E. coli* cell membranes [17]. In the soft *C. glutamicum* cell membranes, MscCG showed a lower activation threshold compared to the channels in *E. coli* cell membranes, implying that differences in bacterial membrane mechanics affect mechanosensitivity of MscCG channels. Since the major lipid component of *C. glutamicum* cell membranes is phosphatidylglycerol (PG) while that of *E. coli* cell membranes is phosphatidylethanolamine (PE) (Table 1), we hypothesized that negatively charged components by PG change membrane stiffness. We visualized liposomal membranes by patch fluorometry and measured bilayer properties using micropipette aspiration. To calculate the areal elasticity modulus K_A_ as an index of membrane stiffness, negative pressure steps up to −50 mmHg were applied to the membranes of the neutral DOPE/DOPC (70%:30%) and negatively charged DOPE/DOPG (70%:30%) liposomes. In excised-patch configuration, the negatively charged membrane DOPE/DOPG (70%:30%) was expanded by pressure much more than the neutral membrane DOPE/DOPC (70%:30%) (Figure 6A, Appendix A). By measuring membrane curvature and membrane surface area, the membrane tension vs. areal strain was plotted. The slope of DOPE/DOPG (70%:30%) was shallower than DOPE/DOPC (70%:30%) (Figure 6B). Moreover, the areal elasticity modulus K_A_ was calculated with the slopes obtained from multiple measurements. The averaged K_A_ of the DOPE/DOPC (70%:30%) and DOPE/DOPG (70%:30%) membranes were 74.9 ± 15.0 (mean ± SEM, *n* = 5) and 30.3 ± 7.9 (mean ± SEM, *n* = 4) mN/m, respectively (Figure 6C). This indicates that negatively charged membranes containing PG are significantly softer than neutral membranes.

## 4. Discussion

In this study, we developed liposomal patch-clamp recording from liposomes fused with native bacterial membrane vesicles (detergent solubilised or non-solubilised) or reconstituted with the purified MscCG channel protein to investigate the “Force-from-lipids” gating of the *C. glutamicum* MS channels. The MscCG-type channels (MscCG and MscCG2) are a well-conserved MscS-like channel subfamily found in *Corynebacteria*. MscCG from *C. glutamicum* that has been shown to function as the major L-glutamate exporter in industrial MSG production. The glutamate export is thought to occur due to increased membrane tension resulting from alterations in membrane lipid composition in bacterial cells grown under biotin limitation, which should specifically activate the MscCG channels. Our results indicate that mechanosensitivity of the MscCG channels is strongly dependent on anionic lipids present in bacterial membranes. Given that most of the MscS-like channels have more than three transmembrane helices compared to the canonical MscS of *E. coli*, and as structurally complex membrane proteins may be involved in more specialized functions rather than functioning only as osmotic safety valves [5,44], it is likely that structurally complex MscS-like channels such as MscCG have evolved by “evolutionary tinkering” [43] to become optimally adapted to sense the mechanical force in specific lipid bilayers of various bacterial membrane environments. This notion seems supported by the fact that plant, algal, and fungal MscS-like channels with more complex structures than canonical MscS, have adapted to sense the mechanical force in eukaryotic organellar membranes, including the membranes of chloroplasts (MSC1 from *Chlamydomonas reinhardtii*, MSL2, and MSL3 from *Arabidopsis thaliana*) [53,54], mitochondria (MSL1 from *Arabidopsis thaliana*) [55] and endoplasmic reticulum (Msy1 and Msy2 from *Schizosaccharomyces pombe*, MscA and MscB from *Aspergillus nidulans*) [56,57].

Liposome reconstitution is a powerful method to investigate lipid effects on the gating of mechanosensitive channels [58,59,60,61]. A worthwhile note here is that due to lack of phosphatidylcholine, blisters from liposomes made of *E. coli* polar lipid extract and synthetic DOPE and DOPG lipids are not as readily forming as the blisters from standard azolectin liposomes used for patch-clamp experiments. Using liposomes made from a variety of lipids, we demonstrated in this study that the activity of the MscCG channel lacks desensitization and/or inactivation and exhibits frequent subconducting states at bacterial physiological membrane potentials (≤−160 mV corresponding to pipette voltage ≥ +160 mV). The presence of strong gating hysteresis exhibited by MscCG was mainly caused by the delayed closing upon the pressure release compared to the opening upon the pressure application. The gating hysteresis of ion channels is understood as a mechanism enabling the channel molecules to memorize active states for a short time as, for example, the tension-modulated KcsA channels [62]. In the case of the MscCG gating, this concept supports the notion that the open-to-closed transition upon pressure release is longer than the closed-to-open transition upon pressure application, since the MscCG channel closing is significantly delayed. The strong current rectification is characteristic of the MscCG channels as also shown in this study in liposomes. It was also observed and reported in bacterial giant spheroplasts [17], indicating that the rectification is an intrinsic property of the channel protein itself. Altogether, our findings suggest that MscCG has gating properties adapted to L-glutamate release once the channel is activated in industrial MSG production.

This study demonstrates that the activation threshold of MscCG is dependent on lipids present in the native bacterial membrane. In liposomes fused with membrane vesicles (50% purified raw *C. glutamicum* membranes and 50% polar azolectin), the activation threshold ratio of CgMscL and MscCG (P_L_/P_CG_) of 2.00 ± 0.13 is close to the value determined in *C. glutamicum* giant spheroplasts of 2.19 ± 0.35 [17]. In contrast, in liposomes reconstituted with the membrane protein extract containing much less of the *C. glutamicum* membranes, the threshold ratio (P_L_/P_CG_) was significantly lower and equal to 1.36 ± 0.07 due to the higher activation threshold of MscCG relative to CgMscL. This indicates that MscCG is less mechanosensitive in liposomes containing low amounts of the *C. glutamicum* lipid components. Furthermore, the purified MscCG protein reconstituted into 100% polar soy azolectin liposomes exhibited a comparatively high activation threshold that was even higher than activation threshold of the co-reconstituted EcMscL channels. In contrast, the high activation threshold of MscCG channels was significantly reduced in *E. coli* liposomes made of the polar lipid extract. *E. coli* polar lipid extract (67% PE, 23.2% PG, 9.8% CA) that has a complex lipid composition [52], which in our study was mimicked using synthetic lipids (70% DOPE, 20% DOPG, 10% CA18:1) to prepare liposomes for MscCG and CgMscL co-reconstitution. The channel recordings obtained from the synthetic liposomes showed that negatively charged lipids, especially phosphatidylglycerol (PG), predominantly affected MscCG mechanosensitivity. It should be noted that the activation threshold ratio between EcMscL and MscCG (P_L_/P_CG_) in *E. coli* polar extract liposomes was higher than that in liposomes made of 70% DOPE, 20% DOPG, 10% CA (18:1), suggesting that liposomes made of synthetic lipids were not completely mimicking the properties of the bilayers made of *E. coli* polar lipid extract. There are possibly other minor constituents affecting MscCG mechanosensitivity in the bacterial membrane of *E. coli*. In fact, bacterial mechanosensitive channels have been shown to be sensitive to membrane thickness, stiffness, and lipid saturation [29,63,64].

How do negatively charged lipids affect mechanosensitivity of bacterial mechanosensitive channels? Anionic lipids PI, PS, PG, and CA are generally present in the inner leaflet of membrane lipid bilayers. Membrane asymmetry resulting from the distribution of these lipids between the bilayer leaflets has been considered one of the major factors contributing to modulation of ion channel mechanosensitivity [65]. MscL of *Mycobacterium tuberculosis* requires PI for its mechanosensitivity in liposomes [66,67]. The closed structure of YnaI channels showed lipid bindings with CA and PS in its large paddle pockets located in the inner leaflet of membrane bilayers [37]. Prototypical KcsA channels sense membrane tension, and their activation requires the presence of negatively charged phospholipids in the inner leaflet [68]. However, more than 80% of *C. glutamicum* cell membrane consists of negatively charged lipids, PI, PG, and CA, and thus the bilayer asymmetry originating from these lipids may not be the only property contributing to channel mechanosensitivity. Alternatively, an increase in negatively charged lipids in the *C. glutamicum* cell membrane may be expected to change MscCG mechanosensitivity. Mutations and overexpression of fatty acid biosynthesis genes significantly alter phospholipid composition in cell membranes [69]. *C. glutamicum* cells overexpressing pgsA2 (phosphatidyl glycerophosphate synthase) and cdsA (CDP-diacylglycerol synthase) genes released constitutively L-glutamate [70], suggesting that in the cell membranes of these *C. glutamicum* cells the threshold of MscCG channels is much lower due to the increased presence of PG and CA negatively charged lipids.

Generally speaking, the force-from-lipids gating of MS channels is interpreted in terms of the continuum model of the transbilayer pressure profile [22,71,72]. This model describes accurately the gating of MscL, which has a relatively simple structure and is mainly affected by the forces of the lipid bilayer as a whole [73]. The complementary “lipid moves first” model expands the force-from-lipid model to molecular level by focusing on specific lipid interactions with MS channel proteins [74,75]. Studies focusing on MscS were instrumental to highlight molecular protein-lipid interactions by showing that hydrophobic pockets present in the MscS channel structure are filled by different numbers of lipids in open and closed states, where membrane tension pulls lipids out of the hydrophobic pockets, allowing the channel to change its conformation from closed to open [32]. To understand the differences observed in the gating of MscCG in liposomes made of different types of lipids reported in this study, it is important to recall that MscL’s threshold of activation of ~70 mmHg was unaffected by different types of lipids used for liposome preparations, whereas MscCG activation threshold strongly depended on the lipid type, with phosphatidylglycerol having the strongest effect. Consequently, this indicates that, like MscS, the effect of force-from-lipids on MscCG occurs mainly at the molecular level through interactions with specific lipids rather than through a global effect of the lipid bilayer as a whole.

It is worth mentioning that stiffening liposomal membranes by adding cholesterol makes MscS harder to open and its activation threshold becomes closer to MscL, whilst the threshold of MscL does not change [29]. An open question is how membrane mechanics affects mechanosensitivity of MscS-like channels by changing membrane stiffness. We demonstrated using patch fluorometry micropipette aspiration that negatively charged membranes are softer than neutral membranes. Supportively, measurement by atomic force microscopy for nano-sized liposomes showed that stiffness of the negatively charged liposome DOPG/DOPC (50:50%) is significantly softer than DOPC (100%) [76]. For large size liposomes, properties of giant unilamellar vesicles (GUVs) were measured by micropipette aspiration. DOPG/DOPC (40%:60%)-GUV was demonstrated to be much softer than DOPC (100%)-GUV by calculating the elastic expansion modulus K_A_ [77,78]. Although the absolute values of K_A_ vary by methods, liposome size, and experimental conditions (salt, sugar, and pH), the trend of soft mechanical properties of negatively charged membranes is consistent. Besides lipid-protein interaction for mechanosensing, we propose that MscCG has a mechanism to sense membrane stiffness as suggested by the experiments investigating the effect of polyunsaturated fatty acids (PUFAs) on the gating of the MscS channels [64].

In summary, the structurally complex *C. glutamicum* MscCG channels were purified and functionally reconstituted into liposomes, and their gating characteristics were investigated by the patch clamp technique. Our data demonstrate that mechanosensitivity of MscCG channels is dependent on membrane stiffness due to negatively charged membrane lipids, primarily PG (phosphatidylglycerol). Other than *Corynebacteria*, an extreme example is presented by the highly anionic bacterial cell membrane of *Staphylococcus aureus* (~90% PG of total membrane lipids [79]). Staphylococcal MS channels are activated by adhesion force (~4 nN) in infectious biofilm formation and antibiotic stress [80,81]. Soft membrane mechanical properties may contribute to the activation of MS channels in response to these specific mechanical cues. The unique structure of MscCG within the MscS-like channel family has become evolutionary, specialized not only to function as an effective L-glutamate exporter, but also to sense mechanical force in the unique corynebacterial cell membrane lipid constitution. Since *C. glutamicum* has been utilized in a broad range of industrial production of amino acids, reconstituted liposome techniques for study of the *C. glutamicum* mechanosensitive channels offer an excellent approach to identify and establish more sustainable and efficient cell factories for amino acid production.

## 5. Conclusions

A series of functional liposome reconstitution techniques of *Corynebacterium glutamicum* MscCG channels was successfully developed. Mechanosensitivity of MscCG channels reconstituted into liposomes depends on the lipid type used for liposome preparations, especially phosphatidylglycerol (PG) as the major lipid determining the MscCG activation threshold in the *C. glutamicum* cell membrane. The *Corynebacterium glutamicum* membranes are very soft due to the PG anionic lipid component. Diverse MscS-like channel structures in bacteria imply evolutionary adjustment of the mechanosensitive channels to the specific force-from-lipids-dependent channel function in the native bacterial membrane.

## Figures and Tables

**Figure 1 microorganisms-11-00194-f001:**
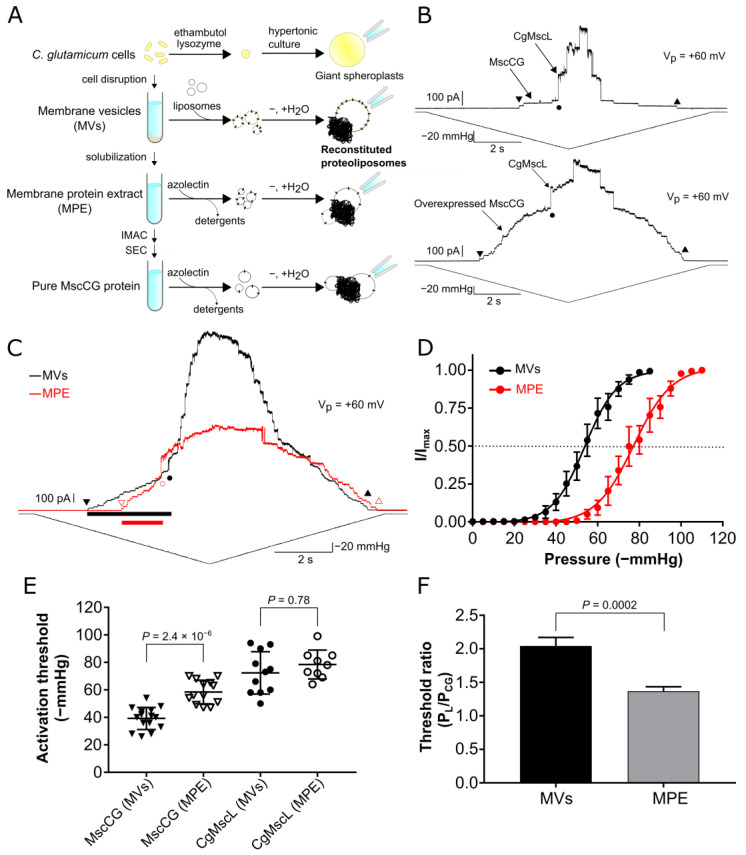
Overexpression of MscCG channels and its mechanosensitivity in liposomes fused with corynebacterial membrane vesicles and liposomes containing membrane protein extract. (**A**) Schematic diagram of liposomal patch-clamp techniques for MscCG mechanosensitive channel purification. (**B**) Current recordings of MscCG and CgMscL in liposomes fused with membrane vesicles. The top trace shows mechanoresponsive currents of MscCG and CgMscL at the native expression level, and bottom trace shows the currents when MscCG was overexpressed. Triangles show the first opening (downward) and the last closing of MscCG (upward), respectively. Circles show the first opening of CgMscL. (**C**) Superimposed traces of mechanosensitive channel currents in liposomes fused with membrane vesicles (MVs) (black) and liposomes containing membrane protein extract (MPE) (red) prepared from MscCG-overexpressing cells. Bars show the distance between the first opening of MscCG (downward closed and open triangles) and CgMscL (closed and open circles). Upward triangles show the last closing of MscCG. (**D**) Boltzmann activation curve of MscCG in liposomes fused with MVs (black) (*n* = 7) and liposomes containing MPE (red) (*n* = 6). Dashed line shows half activation level. (**E**) First opening threshold of MscCG and CgMscL in liposomes fused with MVs and liposomes containing MPE. (**F**) First opening threshold ratio between CgMscL and MscCG (P_L_/P_CG_) in liposomes fused with MVs and liposomes containing MPE. *p*-values were calculated by *t*-test (two-sample assuming unequal variances) to compare between two groups.

**Figure 2 microorganisms-11-00194-f002:**
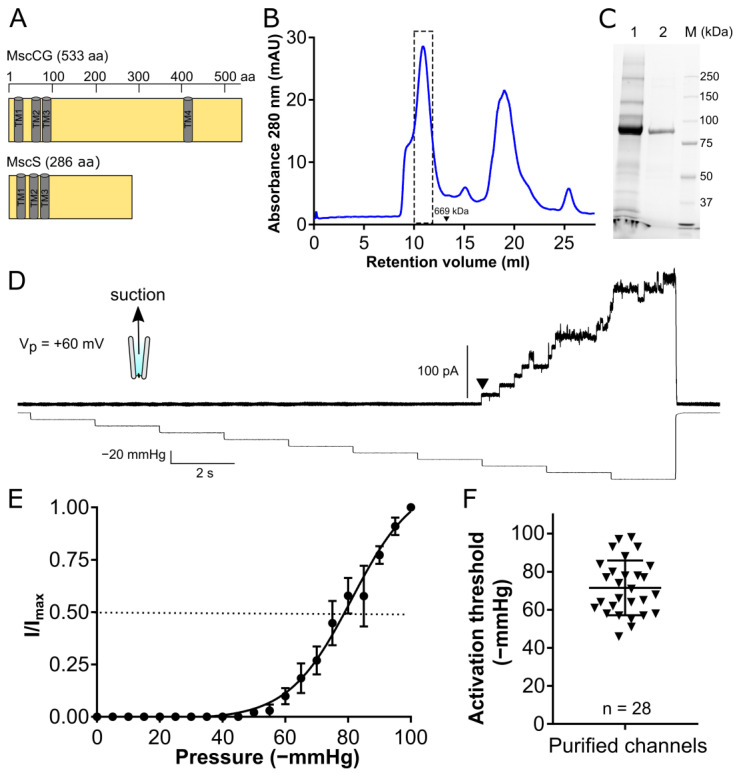
Protein purification and liposomal reconstitution of the MscCG channels. (**A**) Secondary structure comparison between *C. glutamicum* MscCG and *E. coli* MscS. (**B**) Size exclusion chromatography (SEC) profile of the IMAC-purified N-terminal 6 × histidine tagged MscCG. The dashed square shows collected fractions for liposomal reconstitution for patch-clamp. (**C**) SDS-PAGE gel results of the purified N-terminal 6 × histidine MscCG protein. Lane, 1: IMAC elutant, 2: Collected and concentrated SEC 11 mL peak fractions, M: marker (Bio-Rad #1610363). (**D**) Mechanosensitivity of the purified MscCG protein reconstituted in 100% polar azolectin liposomes under pressure steps at +60 mV pipette voltage. The downward triangle shows the first opening of MscCG channel. (**E**) Boltzmann activation curve of the purified MscCG channel (*n* = 7). The dashed line shows half activation level. (**F**) First opening threshold determination of the purified MscCG channel (*n* = 28).

**Figure 3 microorganisms-11-00194-f003:**
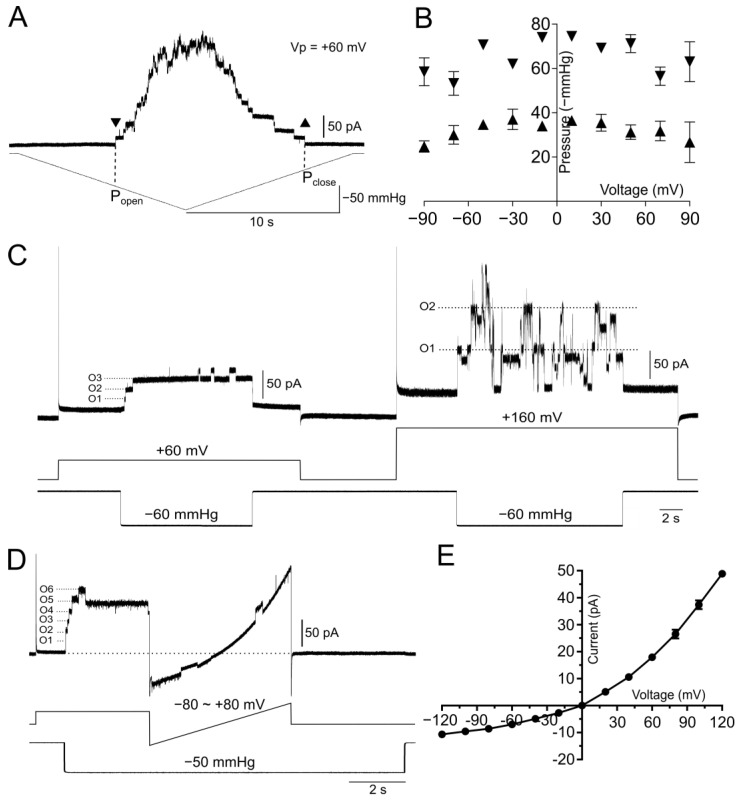
Electrophysiological characterization of the purified MscCG protein reconstituted into liposomes. (**A**) Gating hysteresis of the purified MscCG channels. Triangles show the first opening P_open_ (downward) and the last closing P_close_ (upward) of MscCG channels. (**B**) Voltage dependency of the first opening P_open_ (downward triangles) and the last closing P_close_ (upward triangles). Data were obtained from four independent patches. (**C**) Stable opening of three channels (O1-3) at +60 mV and subconducting states of two open channels (O1-2) at +160 mV under −60 mmHg step. (**D**) Stable opening of channels (O1-6) under −50 mmHg step and current rectification following a voltage ramp from −80 mV to +80 mV. (**E**) Single channel conductance I–V curve of the purified MscCG channels (*n* = 4–11). Bars show standard deviation.

**Figure 4 microorganisms-11-00194-f004:**
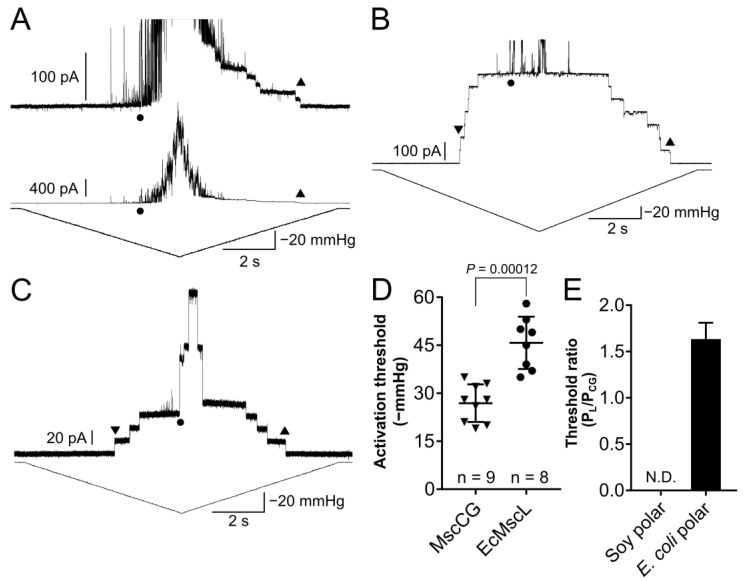
Mechanosensitivity of MscCG and co-reconstituted EcMscL in soy and *E. coli* polar extract liposomes. (**A**) Current recordings of MscCG and EcMscL co-reconstituted in soy polar extract liposomes. Circles show the first opening of EcMscL, and upward triangles show the last closing of MscCG. A zoomed current trace is for small MscCG currents (**top**), and an entire trace is for large EcMscL currents (**bottom**). (**B**) Current recordings of canonical *E. coli* MscS and EcMscL co-reconstituted in soy polar extract liposomes. Circle shows the first opening of EcMscL. Downward and upward triangles show the first opening and the last closing of MscS, respectively. (**C**) Current recordings of MscCG and EcMscL co-reconstituted in *E. coli* polar extract liposomes. Downward and upward triangles show the first opening and the last closing of MscCG, respectively. Circle shows the first opening of EcMscL. (**D**) First opening activation threshold of MscCG and EcMscL in liposomes made of *E. coli* polar lipid extract. Downward triangles show the first opening threshold of MscCG. Circles show the first opening threshold of EcMscL obtained from multiple experiments in (**C**). (**E**) First opening threshold ratio between EcMscL and MscCG (P_L_/P_CG_) in liposomes made of *E. coli* polar lipid extract. *p*-values were calculated by *t*-test (two-sample assuming unequal variances) to compare between two groups.

**Figure 5 microorganisms-11-00194-f005:**
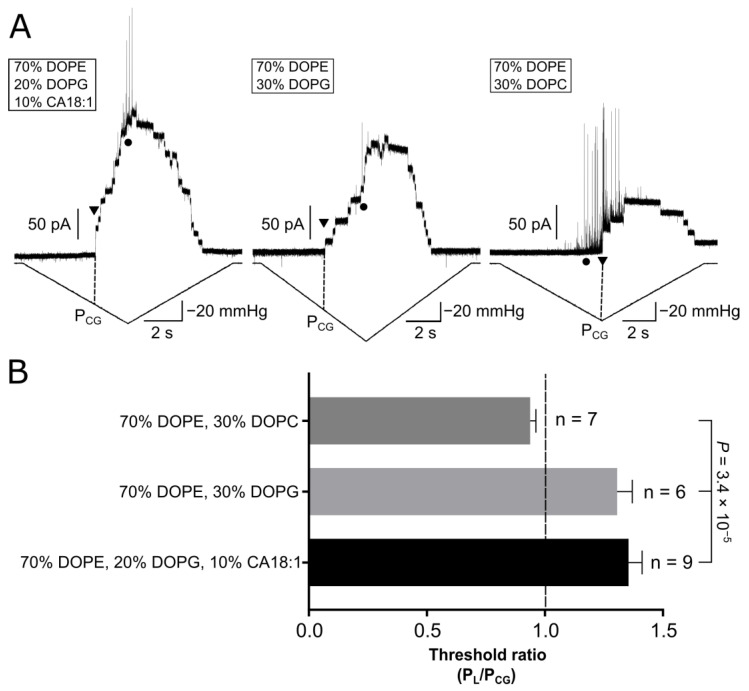
Mechanosensitivity of co-reconstituted MscCG with EcMscL in synthetic lipid liposomes. (**A**) Current recordings of purified MscCG and EcMscL co-reconstituted in liposomes made of different synthetic lipids of 70% DOPE, 20% DOPG, 10% CA18:1 (**left**), 70% DOPE, 30% DOPG (**center**), and 70% DOPE, 30% DOPC (**right**). Downward triangles show the first opening of MscCG, and circles show the first opening of EcMscL. P_CG_ shows pressure level at the first opening activation for MscCG. (**B**) The comparison of the first opening threshold ratio between EcMscL and MscCG (P_L_/P_CG_) in three different liposome preparations made of synthetic lipids. *p*-value was calculated by ANOVA (single factor) among three groups.

**Figure 6 microorganisms-11-00194-f006:**
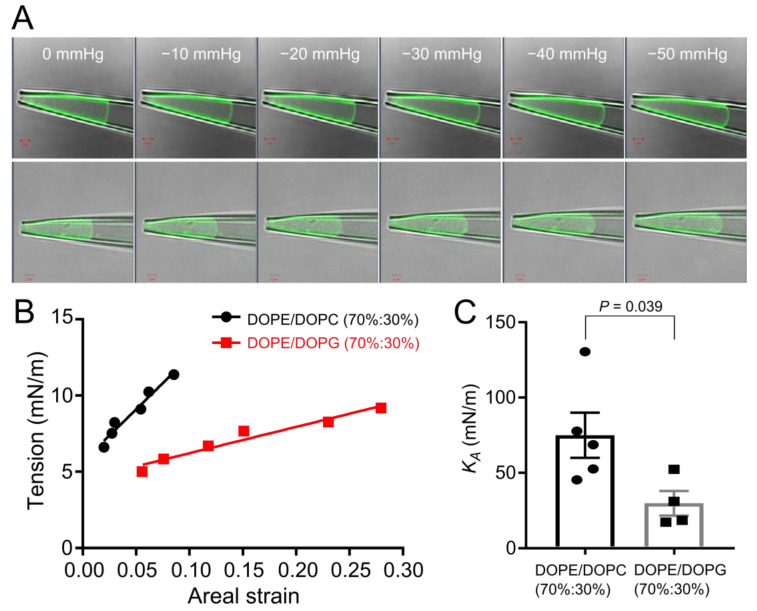
Evaluation of membrane stiffness by micropipette aspiration combined with patch fluorometry. (**A**) Fluorescent images of liposomal excised patch membranes under various pressures made of different synthetic lipids of DOPE/DOPC (70%:30%) (**top**) and DOPE/DOPG (70%:30%) (**bottom**). Scale bars show 2 μm. (**B**) Comparison of the slope of the membrane tension vs. areal strain between DOPE/DOPC (70%:30%) (black circles) and DOPE/DOPG (70%:30%) (red squares). (**C**) The areal elasticity modulus K_A_ of DOPE/DOPC (70%:30%) (circles) (*n* = 5) and DOPE/DOPG (70%:30%) (squares) (*n* = 4). *p*-values were calculated by *t*-test (two-sample assuming unequal variances) to compare between two groups.

**Table 1 microorganisms-11-00194-t001:** Lipid components of soy polar extract, *E. coli* polar extract, and *C. glutamicum* cell membranes. Note that the information about the components of the soy polar lipid and *E. coli* polar lipid extract is from the website of Avanti Polar Lipids, Inc. (https://avantilipids.com/) and the information about the components of *C. glutamicum* cell membranes is cited with the permission of the author from Nagakubo, T. et al. Mycolic acid-containing bacteria trigger distinct types of membrane vesicles through different routes. *iScience*
**2021**, *24*, 102015.

	Soy Polar Lipid Extract	*E. coli* Polar Lipid Extract	*C. glutamicum* Cell Membranes Ref. [51]
Component	wt/wt%	wt/wt%	Relative Abundance (%)
Phosphatidylcholine (PC)	45.7	-	-
Phosphatidylethanolamine (PE)	22.1	67.0	-
Phosphatidylinositol (PI)	18.4	-	6.7
Phosphatidic acid (PA)	6.9	-	-
Phosphatidylglycerol (PG)	-	23.2	72.4
Cardiolipin (CA)	-	9.8	6.5
Diacylglycerol (DAG)		-	14.4
Unknown	6.9	0.0	-

## Data Availability

The analyzed data presented in this study are included within the main article and Appendix A.

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
