# Peer review of "“Force-From-Lipids” Dependence of the MscCG Mechanosensitive Channel Gating on Anionic Membranes"

_microorganisms, 2023, doi:10.3390/microorganisms11010194_

Round 1

Reviewer 1 Report

Review for

 “Force-from-lipids” dependence of the MscCG mechanosensitive channel gating on anionic membranes

A series of functional liposome reconstitution techniques of Corynebacterium glutamicum MscCG channels was successfully developed.

As discussed in the Acknowledgments

Why authors chose such a model?

Acknowledgments: We thank Prof. Hisashi Kawasaki for the discussions about microbial cell factories using Corynebacterium glutamicum

------------------------------

Many papers already published by the same team

Corynebacterium glutamicum mechanosensing: From osmoregulation to L-Glutamate secretion for the avian microbiota-gut-brain axis

Nakayama, Y.    2021       Microorganisms

9(1),201, pp. 1-19

Mechanosensitive channels of Corynebacterium glutamicum functioning as exporters of L-glutamate and other valuable metabolites

Kawasaki, H., Martinac, B.           2020      Current Opinion in Chemical Biology

59, pp. 77-83

“Force-From-Lipids” mechanosensation in Corynebacterium glutamicum

Nakayama, Y., Hashimoto, K.-I., Kawasaki, H., Martinac, B.           2019      Biophysical Reviews

11(3), pp. 327-333

                Evolutionary specialization of MscCG, an MscS-like mechanosensitive channel, in amino acid transport in Corynebacterium glutamicum

Nakayama, Y., Komazawa, K., Bavi, N., (...), Kawasaki, H., Martinac, B.     2018      Scientific Reports

8(1),12893

Etc, etc

This new research brings new elements such as

Mechanosensitivity of MscCG channels reconstituted into liposomes depends on the lipid type used for liposome preparations, especially phosphatidylglycerol (PG) as the major lipid determining the MscCG activation threshold in the C. glutamicum cell membrane.

Corynebacterium glutamicum membranes are very soft due to the PG anionic lipid component. Diverse MscS-like channel structures in bacteria imply evolutionary adjustment of the mechanosensitive channels to the specific force-from-lipids-dependent channel function in the native bacterial membrane.

15 document results

TITLE-ABS-KEY ( msccg  AND  channel ) 

In Scopus search

Mechanism presents in other bacterial genera/species?

General comment: very nice experimental study, well conducted

Discussion supported by strong experimental results

Author Response

We appreciate the reviewer's helpful comments to improve our manuscript. Please find attached a PDF file as our response to the comments.

Reviewer 2 Report

In this study, the authors described a development of liposome reconstitution techniques of Corynebacterium glutamicum MscCG channels. The authors reported the phosphatidylglycerol (PG) in the lipid bilayer determine the mechanosensitivity of MscCG channels.

The study is well-designed and has an acceptable flow

Minor comments

1- Figure 1E: First opening threshold of MscCG and CgMscL in liposomes fused 95 with MVs and liposomes containing MPE. The p value between MscCG (Mvs) and MscCG (MPE) is strange. Please check

2- Figure 4D and 4E: there is lack of statistic between MscCG vs EcMscL. Also between Soy polar and E.coli polar.

3- Table 1:  Soy polar extract: unknown (6.9)?? What is the unknown? Statistic difference between the components are required. Also In the discussion the authors need to discuss the difference in lipid component and how this difference affect the liposome formulation.

Author Response

We appreciate the reviewer for positive comments on our study. Please find attached a PDF file as our response to the comments. 
